# Macrophage Migration Inhibitory Factor (MIF) as a Stress Molecule in Renal Inflammation

**DOI:** 10.3390/ijms23094908

**Published:** 2022-04-28

**Authors:** Yao-Zhong Kong, Qiyan Chen, Hui-Yao Lan

**Affiliations:** 1Nephrology Department, The First People’s Hospital of Foshan, Foshan 528000, China; chenqiyan1@foxmail.com; 2Department of Medicine & Therapeutics, Li Ka Shing Institute of Health Sciences, The Chinese University of Hong Kong, Hong Kong SAR, China

**Keywords:** MIF, inflammation, macrophages, T cells, kidney diseases

## Abstract

Renal inflammation is an initial pathological process during progressive renal injury regardless of the initial cause. Macrophage migration inhibitory factor (MIF) is a truly proinflammatory stress mediator that is highly expressed in a variety of both inflammatory cells and intrinsic kidney cells. MIF is released from the diseased kidney immediately upon stimulation to trigger renal inflammation by activating macrophages and T cells, and promoting the production of proinflammatory cytokines, chemokines, and stress molecules via signaling pathways involving the CD74/CD44 and chemokine receptors CXCR2, CXCR4, and CXCR7 signaling. In addition, MIF can function as a stress molecule to counter-regulate the immunosuppressive effect of glucocorticoid in renal inflammation. Given the critical position of MIF in the upstream inflammatory cascade, this review focuses on the regulatory role and molecular mechanisms of MIF in kidney diseases. The therapeutic potential of targeting MIF signaling to treat kidney diseases is also discussed.

## 1. Introduction

Macrophage migration inhibitory factor (MIF) is a high conserved pleiotropic cytokine which contains 115 amino acids with a molecular weight of 12.5 kDa [1,2]. MIF not only promotes the activation of macrophages and T cells, but also plays a vital role in promoting inflammatory responses including the chemo-attractive effect on immune cells, the production of proinflammatory cytokines and stress molecules, antagonizing the immunosuppressive effect of glucocorticoids [2,3,4]. Therefore, MIF contributes to the pathogenesis of many immune and autoinflammatory diseases including sepsis [5], atherosclerosis and cardiovascular disease [6,7,8], rheumatic arthritis (RA) and systemic lupus erythematosus (SLE) [9], gastric and liver inflammation [10,11], neuroinflammation [12], obesity and diabetes [13], and kidney diseases [14,15].

MIF functions in both autocrine and paracrine manners through binding the receptors CD74/CD44, CXCR2, CXCR4, and CXCR7, which results in the activation of the downstream ERK1/2, AMPK, and AKT signaling to mediate the proinflammatory responses [3]. Furthermore, MIF is induced by glucocorticoids (GCs) and functions as a GC counter-regulator to regulate inflammation and immunity [16,17]. Once released, MIF regulates the GC-induced annexin1 expression and the release of eicosanoids in macrophages [17]. Thus, MIF works together with GC in regulating immune responses under disease conditions. Renal inflammation is an initial step triggering the progression of kidney disease and macrophages are a key cell type that drives the inflammatory process in kidney diseases [18,19,20,21]. Under normal conditions, MIF is expressed at low levels in kidney tissues, but it is significantly upregulated in the injured kidney by immune cells, podocytes, tubular cells, mesangial cells, endothelial cells, and fibroblasts [14,15]. Among them, macrophages are a rich source of MIF during renal inflammation. In this review, we will focus on the immunological functions of MIF in kidney diseases.

## 2. Role of MIF in Acute Kidney Injury

As shown in Figure 1, MIF plays a role in acute kidney injury (AKI) under various diseased conditions including renal ischemia, toxicity, infection and sepsis, acute graft rejection, post-renal obstruction, immunologically mediated GN, and congenital anomalies of the kidney and urinary tract (CAKUT). AKI is defined by acute tubular necrosis accompanied by a rapid increase in serum creatinine while decreasing urine output. Increasing evidence shows that AKI is a common kidney disease and a significant global health concern [22]. The etiology of AKI has been widely explored from multiple perspectives including infection, sepsis, renal ischemia, toxicity, and hypoxia [23]. Severe renal inflammation and oxidative stress including the infiltration of macrophages, T cells, and neutrophils, and the upregulation of proinflammatory cytokines are the key pathological processes of AKI [22,23,24,25,26]. Among proinflammatory cytokines such as MCP-1, IL-1β, TNFα, and IL-6, MIF has been shown to be a risk factor of AKI. Indeed, severe renal inflammatory responses may stimulate a rapid release of MIF, resulting in high serum and urinary levels of MIF. In addition, impairments of the glomerular filtration rate and urinary output in AKI patients may be other factors contributing to elevated levels of MIF. Thus, urinary and plasma MIF levels are associated with the development of AKI in patients with acute pyelonephritis [27], severe sepsis and septic shock [28], after cardiac surgery or liver transplantation [29]. A recent study also found that plasma and urinary MIF levels are largely elevated at the onset of AKI but decline to normal levels when AKI is resolved [30]. In addition, both serum and urinary levels of MIF are closely correlated with an increase in serum creatinine, regardless of what caused the disease in patients with AKI [30]. In critically ill patients, high serum MIF levels predict the development of AKI and mortality [31,32]. Experimentally, increased MIF expression is associated with AKI in acute renal allograft rejection [33] and renal ischemia/reperfusion (I/R)-induced AKI [30,34]. The pathogenic role of MIF in AKI is uncovered by recent studies that mice with genetic deletion or pharmacological inhibition of MIF are protected from IRI or cisplatin-induced AKI [30,35]. Interestingly, a long-noncoding RNA (LRNA9884)-mediated cisplatin-induced AKI is associated with the NF-kB-mediated transcriptional activation of MIF [36]. All these studies delineate a pathogenic role in AKI.

However, it should be pointed out that MIF may play a diverse role in AKI (Figure 2). In hypoxia or under severe renal stress conditions, a large amount of MIF is rapidly released and thus triggers severe renal inflammation. Under such conditions, MIF may be pathogenic and play an early role in orchestrating the initial cellular response to tissue injury. Indeed, hypoxia can cause MIF to be rapidly released from pre-formed intracellular pools to trigger the inflammatory response including the expression of MCP-1, TNF-α, IL-1β, IL-6, iNOS, CXCL15(IL-8 in human) and the recruitment and activation of macrophages, neutrophil, and T cells, resulting in severe AKI [31,32]. However, MIF may also play a reparative role in AKI by promoting tubular cell proliferation while inhibiting apoptosis or cell cycle arrest if MIF levels are not sufficiently high to trigger severe renal inflammation. Under this situation, MIF may be protective in AKI as demonstrated in recent studies that mice lacking MIF develop worse AKI by inhibiting tubular epithelial cell proliferation [37,38]. Thus, renal microenvironments may influence the role of MIF. It is highly possible that MIF at higher concentrations may cause severe renal inflammation to mediate AKI as reported in many studies of patients and animal models with progressive renal injury [26,27,28,29,30,31,32,33,34,35]. It is also possible that under certain disease conditions such as unilateral IRI-induced AKI in which AKI is induced by the ligation of one renal artery only [37], MIF may be protective by promoting cell proliferation during the repair process without inducing severe renal inflammation. Similarly, MIF also promotes tubular epithelial proliferation to limit renal inflammation and fibrosis by counteracting tubular cell cycle arrest [38,39]. Nevertheless, the pathogenic role of MIF in AKI warrants further investigation.

## 3. Role of MIF in Immunologically Mediated Kidney Diseases

### 3.1. MIF in Crescentic Glomerulonephritis (GN)

MIF has been shown to play a critical role in chronic kidney disease (CKD, Figure 1). Among them, crescentic glomerulonephritis (GN) is a severe form of glomerular injury characterized by the disruption of the glomerular basement membrane, cellular proliferation and macrophage accumulation within Bowman space accompanied by fibrinoid necrosis. Previous studies showed that serum and urinary MIF is highly elevated in CKD patients with crescentic GN and in experimental anti-GBM crescentic GN [40,41,42,43,44]. In both human and experimental GN, MIF is highly expressed by intrinsic kidney cells and infiltrating macrophages and T cells [40,41], which may contribute to high levels of serum and urinary MIF. The pathogenic role of MIF in crescentic GN is demonstrated in a mouse model of crescentic GN in which mice lacking MIF and its receptor CD74 are protected from the development of glomerular crescents [44]. Further studies by blocking MIF with neutralizing antibodies or an inhibitor also show that the blockade of MIF attenuates both rat and mice models of anti-GBM crescentic GN [45,46,47]. In contrast, specifically podocyte-overexpressing MIF results in progressive glomerulosclerosis and end-stage renal failure [48]. Thus, MIF plays a pathogenic role in crescentic GN.

### 3.2. MIF in Renal Transplantation

Renal allograft rejection is a T cell-dependent process in which the graft becomes highly inflamed when large numbers of T cells and macrophages infiltrate the kidney. T cells can cause graft injury through cytotoxic mechanisms and indirectly via the recruitment and activation of macrophages [49]. MIF is a well-known cytokine associated with the activation of both the innate and adaptive immune system during renal allograft rejection. It has been shown that MIF is produced by both local kidney cells and infiltrating macrophages and T cells in both human and rat models of acute renal allografts [33,50]. Increased MIF expression in allograft rejection also gives a highly significant correlation with macrophage and T cell accumulation and the severity of allograft rejection, and the loss of renal function [50]. Clinically, urinary and plasma levels of MIF predict the renal graft functions [51,52]. It is also reported that genetic polymorphisms of MIF and B-cell activating factor (BAFF) are at risk of the posttransplant development of donor-specific antibodies-mediated renal allograft rejection [53]. In addition, high levels of serum MIF predict the development of chronic allograft nephropathy [54]. However, the blockade of MIF using a neutralizing antibody or by genetically deleting MIF reduces the delayed-type hypersensitivity response without preventing acute renal allograft rejection [55], which warrants further investigation for the pathogenic role of MIF in renal allograft rejection.

### 3.3. MIF in Lupus Nephritis

Lupus nephritis (LN) is another autoimmune kidney disease caused by systemic lupus erythematosus (SLE). In patients with active SLE, elevated serum and urinary MIF levels are detected in lupus patients and positively correlated with the severity of LN [56,57,58]. In lupus-prone mouse strains, the upregulation of MIF and CD74 was also positively correlated with worsening renal inflammation, whereas lupus mice lacking the MIF or CD74 receptor are protected from LN, demonstrating the role of the MIF-CD74 axis in the pathogenesis of LN [59,60]. Further studies show that the blockade of MIF with a neutralizing MIF antibody, small molecule, or a tolerogenic peptide (hCDR1) can also improve renal dysfunction and reduce leukocyte recruitment and inflammatory cytokine production [61,62], confirming the pathogenic role of MIF in LN.

### 3.4. MIF in IgA nEphropathy

It has been reported that urinary MIF is increased in patients with IgA nephropathy (IgAN) and correlated with progressive renal injury including glomerular crescent formation [15,42,63,64]. MIF produced by T cells from IgAN patients was also reduced after steroid treatment [64], suggesting the counter-regulation between MIF and glucocorticoids in the pathogenesis of IgA nephropathy. In vitro, polymeric IgA isolated from patients with IgAN is capable of inducing MIF production in cultured human mesangial cells [65]. In experimental IgA nephropathy, treatment with a neutralizing MIF antibody can inhibit IgAN by blocking TGF-β1 expression [66], suggesting a role of MIF in the pathogenesis of IgAN.

## 4. Role of MIF in Other Kidney Diseases

### 4.1. MIF in Diabetic Nephropathy

Macrophages are a major inflammatory cell in diabetic nephropathy (DN) [67,68], which may be associated with the upregulation of MIF. It has been shown that both MIF and CD74 in serum and urine are dramatically elevated in patients with type 2 diabetes and are positively correlated with severe podocyte injury [69]. Indeed, MIF is increased in podocytes and tubular cells from humans and animals with DN and is capable of inducing MCP-1 and TRAIL (TNF-related apoptosis inducing ligand) expression by podocytes and tubular cells [70]. Interestingly, the treatment of type 2 diabetes with metformin results in a significant fall in urinary MIF and CD74 excretion, suggesting that the anti-inflammation mechanism of metformin reduces podocyte injury and albuminuria by suppressing MIF-CD74-mediated renal inflammation [69]. Furthermore, the blockade of MIF ameliorated functional and histopathological injury in the DN rats and db/db mice by lowering blood glucose, and inhibiting albuminuria, extracellular matrix accumulation, epithelial–mesenchymal transition (EMT), and macrophage activation [71,72]. Thus, MIF may play a pathological role in diabetic nephropathy.

### 4.2. MIF in Autosomal Dominant Polycystic Kidney Disease (ADPKD)

ADPKD is characterized by renal cyst formation, inflammation, and fibrosis, and is associated with renal interstitial inflammation [73]. Recently, several studies have shown that MIF plays a critical role in ADPKD. Urinary MIF is increased in patients with ADPKD [15,43]. HIF-1α and cAMP can induce the expression of MIF by primary tubular cells and cyst-lining epithelial cells and promotes cyst cell proliferation independently of macrophages [74]. Further study also found that MIF promotes cystic epithelial cell proliferation by activating ERK, mTOR, and Rb/E2F pathways, and regulates cystic renal epithelial cell apoptosis through p53-dependent signaling [74]. Functionally, MIF deletion or blockade can delay the cyst growth in ADPKD mice [75], indicating that MIF may involve in the pathogenesis of ADPKD.

## 5. Mechanisms of MIF in Kidney Disease

### 5.1. MIF Receptors and Signal Pathways in Renal Inflammation

MIF is an upstream cytokine in the inflammatory cascade and is released upon stimulation by cellular stress, endotoxin, exotoxin, infection, inflammation, and immune responses. Once released, MIF acts as a proinflammatory cytokine to induce the expression of other inflammatory cytokines/mediators including IL-1, TNF-α, IL-2, IL-6, IL-8, INF-γ, and iNOS to further promote renal inflammation and immune responses by binding to its receptors [14] (Figure 3). It is well established that MIF exerts its biological functions in an autocrine and paracrine manner via the CD74, CD44, CXCR2, CXCR4, and CXCR7 receptors [76,77,78,79,80,81]. As illustrated in Figure 3, it is also known that the binding of MIF to receptor CD74 to initiate downstream signaling requires the recruitment of CD44 or CXCR receptors [77], including the CD74/CD44 [77,78], CD74/CXCR2 [79], CD74/CXCR4 [80], and CD74/CXCR4/CXCR7 [81]. Although it is not clear whether CD44 is involved in the receptor complexes of CD74 with the CXCRs, the induction of MIF signaling solely via CXCR7 has been reported [82]. Once released, MIF binds receptors to trigger the activation of the downstream signaling pathways including ERK1/2, AMPK, and AKT to exert its bioactivates [3,83]. It is also reported that MIF can induce the activation of the JNK pathway on T cells and fibroblasts via CXCR4 and CD74 [84].

In the kidney, CD74 transduces MIF signals in podocytes, tubular cells, and infiltrating inflammatory cells including macrophages and T cells [70]. Once released, MIF binds CD74 to initiate the membrane recruitment of CD44, resulting in the activation of the downstream signal transduction [71,72]. It has been shown that CD74 is markedly upregulated in a diseased kidney with AKI and CKD and MIF mediates acute and chronic injury via CD74/CD44-ERK1/2 or CD74/TLR4-NF-κB-dependent mechanisms [30,35,44,85,86]. In proliferative GN, the upregulation of the podocyte MIF can induce the proliferation of parietal epithelial cells and mesangial cells via the activation of CD74/CD44 signaling [44]. MIF can also induce integrin-β1 and cyclin D1 expression via the ERK pathway to promote cell proliferation and differentiation. These studies indicate a crosstalk between podocytes and parietal epithelial cells via MIF signaling. It was also reported that MIF is the direct target gene of HIF-1α in human primary tubular cells. The tubule-specific knockout of HIF-1α can inhibit MIF upregulation [74]. Furthermore, MIF is also regulated by cAMP signaling to promote cyst growth in ADPKD [74].

### 5.2. MIF in T Cell-Mediated Kidney Disease

MIF may have a direct or indirect role in recruiting T cells to sites of immune and inflammatory injury as MIF can directly and indirectly activate T cells by inducing the expression of chemokines and adhesion molecules. This is supported by the findings that MIF-producing T cells are exclusively localized to the area of severe tissue injury, including crescentic GN [40,41], IgA nephropathy [41,64], focal glomerular and tubulointerstitial lesions [41], and necrotic vascular inflammation in human renal allograft rejection [50]. MIF may also act by stimulating T cell proliferation and activation to mediate renal injury by promoting the delayed-type hypersensitivity (DTH) and Th1/Th17 immune responses (Figure 2). Indeed, MIF is the first T cell cytokine-associated DTH response. Direct evidence for a role of MIF in the DTH response associated with kidney disease comes from the findings that treatment with a neutralizing anti-MIF antibody inhibits skin DTH reaction in the primed mouse model of anti-GBM crescentic GN [45]. Furthermore, MIF can promote Th1/Th2/Th17 inflammatory responses in human primary cell cultures of PBMC from active SLE patients [87]. The absence of MIF results in obesity and inflammation due to the increase in Treg cells in the visceral adipose tissue of MIF-deficient mice, indicating MIF is a new regulator of Treg cells 7 [88]. Evidence of MIF in T cell-mediated kidney disease comes from the observation that T cell-mediated renal injury is prevented in lupus-prone mice targeted for the deletion of MIF [60], whereas treatment with anti-MIF antibody protects against macrophages and T cell-mediated anti-GBM crescentic GN [45].

### 5.3. MIF in Macrophage-Mediated Kidney Diseases

Macrophages are a rich source of MIF production in the diseased kidney. We find that proinflammatory macrophages produce abundant MIF in both experimental and human kidney disease, including renal allograft rejection [40,41,45,46,50]. In addition, the upregulation of MIF is strongly associated with macrophage accumulation, the severity of tissue injury, and the development of AKI [30,31,32,33,34,35] and crescentic GN [40,41,45,46]. These findings suggest that the local production of MIF by macrophages may, in turn, activate macrophages to produce cytokines (IL-1, TNF-α, IL-2, INF-γ), chemokines (MCP-1), adhesion and oxidative molecules (Figure 2). Evidence to support a critical role of MIF in macrophage-mediated renal injury also comes from the finding that blockade of MIF with a neutralizing MIF antibody is able to prevent or reverse macrophage accumulation and activation in a mouse or rat model of crescentic glomerulonephritis [35,45,46,47].

### 5.4. MIF as Glucocorticoid Antagonist in Renal Injury

MIF has a unique relationship with glucocorticoids as MIF is also secreted from corticotropic anterior pituitary cells together with ACTH that can stimulate adrenal glucocorticoid secretion. As shown in Figure 4, under stress and inflammation conditions, MIF is induced by glucocorticoids but acts as an antagonist of glucocorticoid actions within the immune system to override the immunosuppressive effects of glucocorticoids [4]. MIF overcomes the inhibitory effects of glucocorticoids on TNFα, IL-1β, IL-6, and IL-8 production by LPS-stimulated monocytes in vitro and suppresses the protective effects of steroids against lethal endotoxemia in vivo [4]. Thus, MIF plays a critical role in the host control of inflammation and immunity. MIF-induced renal injury may also be associated with an antagonistic action upon the anti-inflammatory and immunosuppressive effects of glucocorticoids.

Glucocorticoids (GCs) are the first-line treatment regimens for most immunologically mediated kidney diseases, including renal transplantation rejection. However, steroid resistance occurs in approximately 20% patients who are at risk pf progression to end-stage kidney disease [89]. Thus, it is possible that targeting MIF may offer better therapeutic benefits in patients with steroid resistance. By using a comprehensive cytokine analysis in children with idiopathic nephrotic syndrome, MIF plasma levels are increased in patients with steroid resistance and can therefore predict the therapeutic response to GCs [90]. Furthermore, genetic studies have also found MIF -173G/C gene polymorphism in children with steroid resistance [91,92]. The reversal of anti-GBM crescentic GN by blocking MIF is also associated with increasing endogenous GC sensitivity [46]. Thus, MIF may play a pathogenic role in GN by counter-regulating the immunosuppressive effect of GCs.

## 6. MIF as a Therapeutic Target for Kidney Disease

Studies in mice and in humans demonstrate the therapeutic potential of MIF inhibition for kidney diseases. There are several MIF-directed pharmacologic therapeutics including small molecule inhibitors, monoclonal antibodies and nanobodies, and peptide inhibitors, which has been well reviewed in elsewhere [9,93,94], and is outlined in Figure 5.

There are two potential therapeutic options for kidney diseases in terms of targeting MIF: the antibody-based and small molecule inhibitors or antagonists. Both have been shown to have a therapeutic effect on the different types of kidney diseases.

### 6.1. Antibody-Based Therapy for Kidney Diseases

The development of the neutralizing MIF antibody provided the first evidence of anti-MIF treatment in kidney diseases. In anti-GBM crescentic GN, the administration of the anti-MIF monoclonal antibody immediately after disease induction or at day 7 when the established anti-GBM crescentic GN can attenuate the macrophage and T cell-mediated progressive renal injury, including crescent formation and rapidly renal dysfunctions in a rat model [45,46]. In experimental IgA nephropathy, treatment with an anti-MIF monoclonal antibody is also able to suppress renal injury by inhibiting renal TGF-β1 expression [66]. Interestingly, anti-MIF treatment with a neutralizing antibody can inhibit the skin DTH response without protection against renal allograft rejection [55]. The differential effect of anti-MIF treatment on skin DTH response and acute renal allograft rejection may be associated with the alternative chemokines and cytokines released by a highly active acute renal allograft rejection site to compensate for the inhibitory effect of MIF during acute graft rejection. Thus, treatment with anti-MIF antibody may be disease-type dependent.

BaxB01 is a fully human monoclonal antibody targeting a disease-related immunologically distinct isoform of MIF, namely oxidized MIF (oxMIF) [95]. BaxB01 could bind to oxMIF with high affinity to reduce macrophage migration in vitro, and to produce a favorable curative effect on glomerulonephritis [95]. A single administration of BaxB01 can significantly reduce proteinuria and diminish histopathological glomerular crescent formation without signs of systemic toxicity or a negative impact on kidney function [96]. A phase I study (NCT01765790) of imalumab (BAX69), a fully human recombinant antioxidized macrophage migration inhibitory factor antibody, has been launched [97]. This study assesses the safety, tolerability, pharmacokinetics, and antitumor activity of imalumab. However, no clinical trial for anti-MIF antibody treatment for kidney disease has been performed.

### 6.2. Treatment of Kidney Diseases with MIF Inhibitors

#### 6.2.1. The Methyl Ester of (S, R)-3-(4-Hydroxyphenyl)-4,5-Dihydro-5-Isoxazole Acetic Acid (ISO-1)

ISO-1 is the first MIF inhibitor and has been well studied in several experimental kidney diseases. ISO-1 binds the MIF tautomerase active site and inhibits downstream MIF signaling [98]. The oral administration of ISO-1 into two distinct models of SLE, the NZB/NZW F1 and the MRL/lpr mouse strains, can block the interaction between MIF and CD74, resulting in the inhibition of CD74^+^ and CXCR4^+^ leukocyte infiltration, proinflammatory cytokine and chemokine expression, and progressive renal injury in lupus glomerulonephritis [61]. Treatment with ISO-1 in type-2 diabetic db/db mice can also significantly decrease blood glucose, albuminuria, extracellular matrix accumulation, epithelial–mesenchymal transition (EMT), and macrophage infiltration in the diabetic kidney [72]. Furthermore, ISO-1 can also protect against experimental AKI by inhibiting the NLRP3 inflammasome signaling pathway and cell pyroptosis [99,100,101]. These data highlight the feasibility of targeting the MIF-MIF receptor interaction by small-molecule antagonism and support the therapeutic value by targeting MIF in kidney diseases.

#### 6.2.2. Ribosomal Protein S19 (RPS19)

RPS19 is a component of the 40S small ribosomal subunit and binds MIF to block the interaction between MIF and CD74. It has been reported that RPS19 treatment largely prevents the development of anti-GBM crescentic GN by suppressing glomerular crescent formation, glomerular necrosis, and progressive renal dysfunction via mechanisms associated with inactivating MIF-induced ERK and NF-κB signaling, thereby inhibiting macrophage and T cell infiltration as well as Th1 and Th17 responses [47]. Further study also shows that treatment with RPS19 is capable of attenuating cisplatin-induced AKI by inhibiting MIF/CD74/NF-κB-mediated renal inflammation, which includes suppressing TNF-α and MCP-1 expression and the infiltration of F4/80^+^ macrophages, neutrophils, and CD3^+^ T cells in the AKI kidney [47].

#### 6.2.3. Other MIF Inhibitors

Recently, several MIF inhibitors/antagonists have been developed and have been shown to have therapeutic effects on several experimental disease models including diabetes [71,102], bone disease [103,104], and cancer [105]. The blockade of MIF with an MIF antagonist p425 has been shown to significantly decrease urine protein and urine protein/creatinine ratio, serum BUN and creatinine in the streptozotocin-induced diabetic rats [71]. The oral administration of a small-molecule MIF antagonist, CPSI-1306, can also significantly lower blood glucose levels and inhibit proinflammatory cytokines IL-6 and TNF-α expression in a mouse model of streptozotocin-induced diabetes [102]. In addition, other MIF inhibitors including 4-IPP and Chicago sky blue 6B (CSB6B) have also been reported to suppress MIF-induced osteoclastogenesis and osteosarcoma tumorigenesis by targeting NF-κB signaling [103,104,105].

## 7. Conclusions and Perspectives

Together, MIF is a proinflammatory cytokine and stress molecule which plays a role in immunologically and non-immunologically mediated kidney diseases including AKI and CKD. MIF is rapidly released from the injured kidney in response to the stimulations under various disease conditions. Once released, MIF can activate the downstream signaling pathways including ERK/p38/JNK MAPK, PI3/AKT, and NF-κB signaling via receptors of CD74, CD44, and CXCR2/4/7, resulting in the upregulation of proinflammatory cytokines/chemokines/adhesion molecules and the recruitment and activation of macrophages and T cells to cause progressive AKI and CKD. Interestingly, MIF can also be induced by GCs under stress conditions but functions as an antagonist to GCs to counter-regulate the immunosuppressive effect of GCs on renal inflammation. Thus, targeting MIF with anti-MIF antibodies or small molecule antagonists MIF may represent a promising therapeutic approach for the treatment of kidney diseases. Unfortunately, the current MIF-targeted treatments for kidney diseases are largely experimental with few ongoing clinical trials, despite this warranting further investigation.

Recent studies have also found that patients with a polymorphism of the MIF gene including 173G/C [106,107], rs3063368 [108], rs755622 [11,109] may be genetically susceptible to the development of AKI and CKD. Based on these findings, it is likely that patients with MIF gene polymorphisms may be at high risk of developing AKI or CKD, or AKI to CKD progression. These findings also raise a possibility that targeting MIF genetic polymorphisms may be a novel and specific therapy for kidney diseases. Thus, research into the genetic polymorphisms of MIF is challenging and the development of new treatment for kidney diseases by genetically and epigenetically targeting MIF may represent promising avenues of future study.

## Figures and Tables

**Figure 1 ijms-23-04908-f001:**
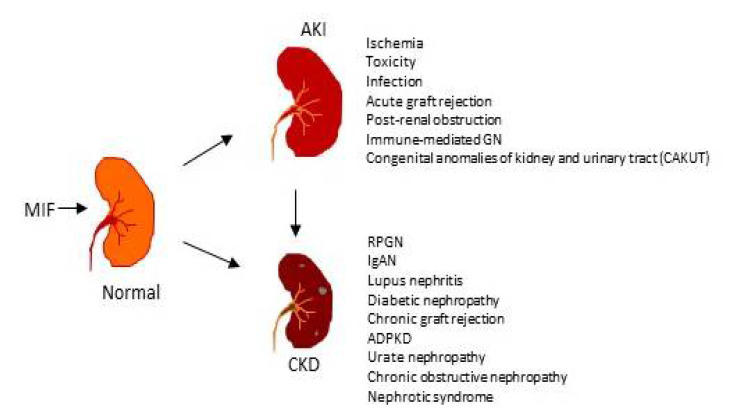
Role of MIF in acute and chronic kidney diseases. Increased exogenous and endogenous MIF can cause AKI and promote AKI-to-CKD, and CKD progression under various disease conditions.

**Figure 2 ijms-23-04908-f002:**
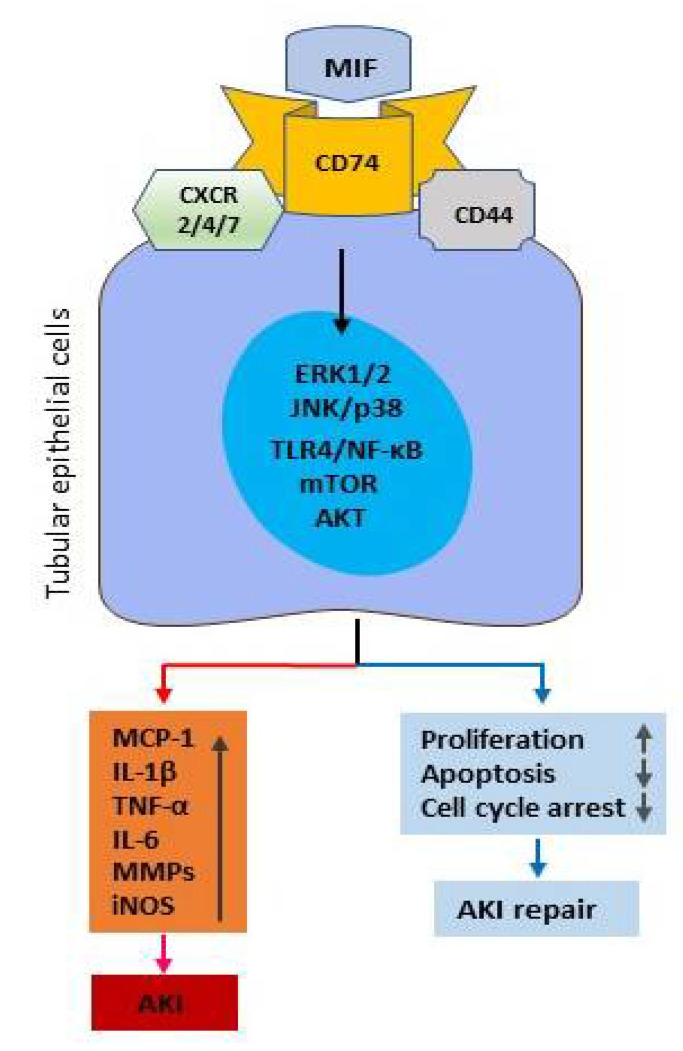
Diverse roles of MIF in AKI. Under certain disease conditions, the overproduction of MIF can promote tubular epithelial cell injury via the proinflammatory mechanisms (the left panel). In contrast, MIF may also activate the downstream pathways to protect tubular epithelial cells from injury by promoting cell proliferation while inhibiting apoptosis and cell cycle arrest (the right panel).

**Figure 3 ijms-23-04908-f003:**
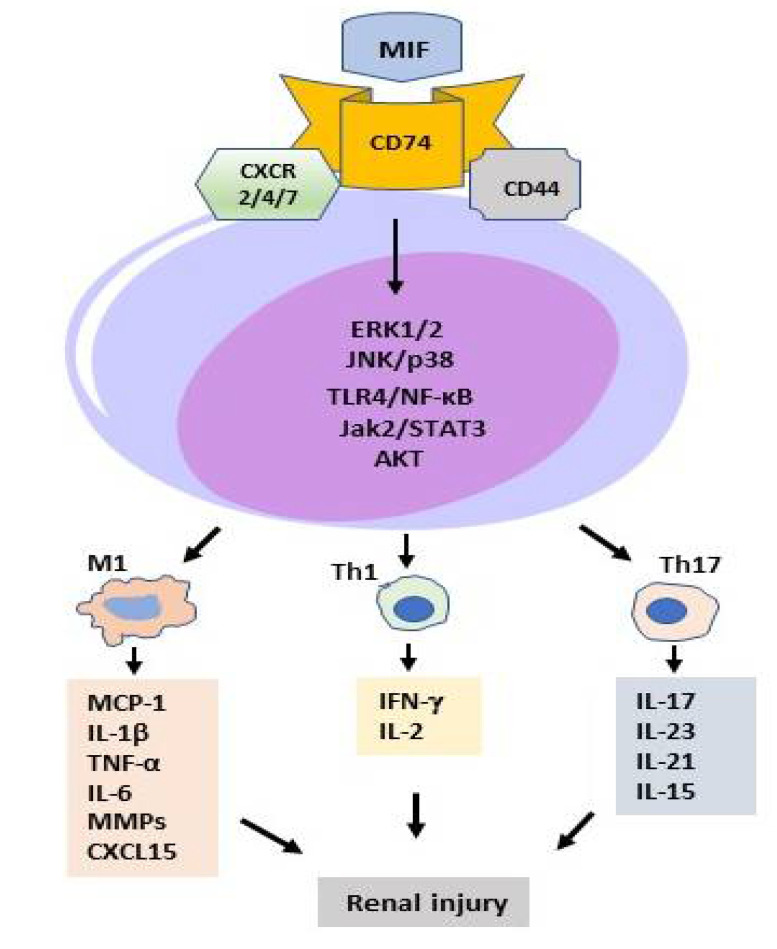
MIF signaling pathways in kidney diseases. After binding to CD74 and CXCRs receptors, MIF can activate the downstream signaling pathways to promote renal inflammation by activating the proinflammatory M1 macrophages and Th1/Th17 immune responses.

**Figure 4 ijms-23-04908-f004:**
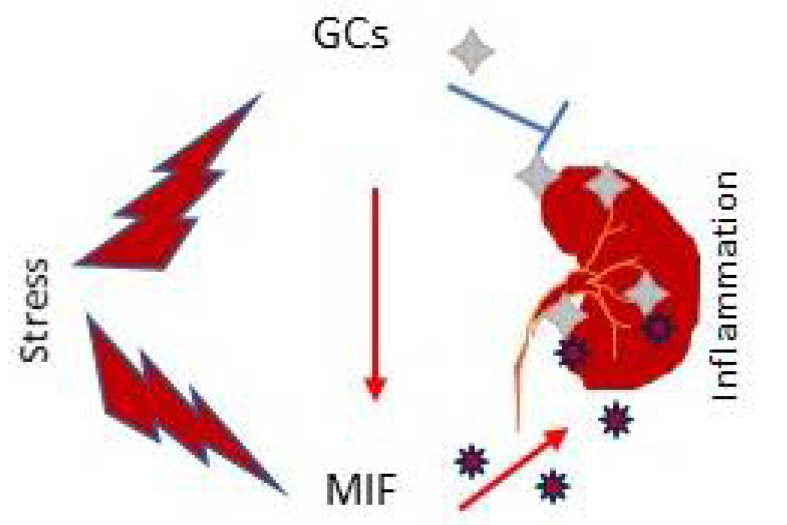
Counter-regulation between MIF and glucocorticoids (GCs) in renal inflammation. Note that MIF and GCs are immediately released in response to the stress stimulation. Once released, GCs can induce MIF to counter-regulate the immunosuppressive actions of GCs, resulting in renal injury.

**Figure 5 ijms-23-04908-f005:**
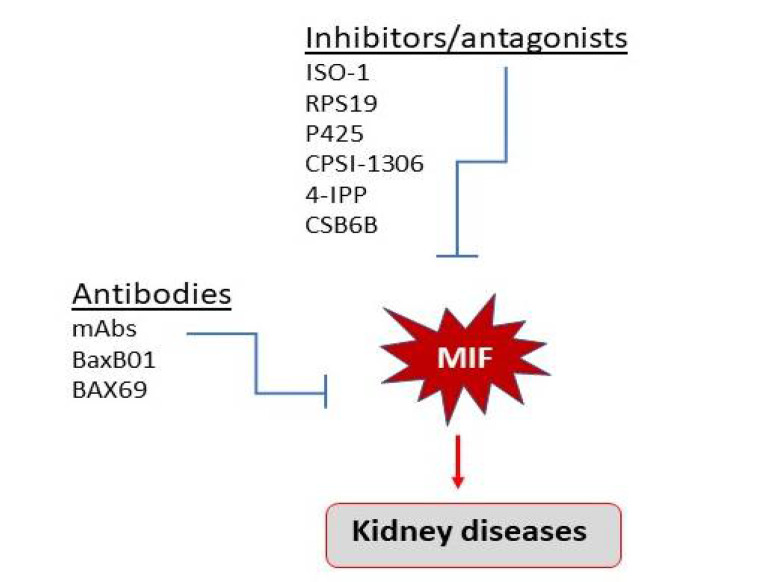
Therapeutic potential for kidney disease by targeting MIF.

## Data Availability

Not applicable.

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
