# Peer review of "Macrophage Migration Inhibitory Factor (MIF) as a Stress Molecule in Renal Inflammation"

_ijms, 2022, doi:10.3390/ijms23094908_

Round 1

Reviewer 1 Report

Comments to the Author

This review is focused on regulatory role of  macrophage inhibitory factor (MIF) and molecular mechanisms in kidney diseases. The authors reviewed this hot topic findings on the contributions of MIF in the biomedical field, with influence on many immune and autoinflammatory diseases. The review seems to be systemic and timely, will provide a go-for resource for researchers in this area when people are looking for a nice overview and analysis of the field. Thus, I recommend to accept this review after some minor revisions.

  1. The review has merely three figures as systemic scheme, without specific figure/schemes to show several individually important stories. This review suggests addition 2-3 figures for important milestone literature stories.
  2. The nanobodies were used to directed pharmacologic therapeutics, you refer that in page 8 line 276. However the authors not include some works or an examples of nanobodies used for kidney diseases.  
  3. The format of references should keep consistent. Please check the reference 46.

  4. In the final conclusion section, limitations and current challenges of this research field should be discussed in a meaningful manner.

Author Response

This review is focused on regulatory role of macrophage inhibitory factor (MIF) and molecular mechanisms in kidney diseases. The authors reviewed this hot topic findings on the contributions of MIF in the biomedical field, with influence on many immune and autoinflammatory diseases. The review seems to be systemic and timely, will provide a go-for resource for researchers in this area when people are looking for a nice overview and analysis of the field. Thus, I recommend to accept this review after some minor revisions.

Response: Thank you very much for the reviewer 1 for the recognition and encouragement of this work.

  1. The review has merely three figures as systemic scheme, without specific figure/schemes to show several individually important stories. This review suggests addition 2-3 figures for important milestone literature stories.

Response:We would like to thank the reviewer 1 for this insightful and structural suggestion. As suggested by the reviewer, we have substantially revised the manuscript by including total of 5 scheme Figures, including the role of MIF in AKI and CKD (Figure 1), the diverse roles of MIF in tubular injury in AKI (Figure 2), the signaling mechanisms of MIF in macrophage and T cell immune responses (Figure 3), the counter-regulation of MIF and glucocorticoids in renal inflammation (Figure 4), and the therapeutic potential by targeting MIF (Figure 5).

  1. The nanobodies were used to directed pharmacologic therapeutics, you refer that in page 8 line 276. However the authors not include some works or an examples of nanobodies used for kidney diseases.

Response: as suggested by the reviewer 1, a new reference (Ref. 93) for anti-MIF treatment with nanobody has been included in the revised manuscript.

  1. The format of references should keep consistent. Please check the reference 46.

Response: Thank you for pointing out this, we have carefully checked all the references to meet the journal reference style.

  1. In the final conclusion section, limitations and current challenges of this research field should be discussed in a meaningful manner.

Response: As suggested by the Reviewer 1, we have revised the conclusion to highlight the current understanding of MIF in AKI and CKD. We have also discussed the limitations and current challenges of MIF in kidney disease in a more meaningful manner. The MIF polymorphisms and the future challenges are also highlighted and discussed in the perspective section. All changes in this section are shown in RED front.

In general, I think this review demonstrated a nice summary of current theraupeutic strategies kidney diseases, and it is helpful for researchers who work or plan to work in this field. There are a few detailed comments and suggestions for the authors.
Overall I thought it was an interesting experimental paper and if it is completed with what I have suggested its potential will be even greater and published.

Response: Thank you for the encouragement.

Reviewer 2 Report

The subject is potentially interesting and the review contains valuable data explaining the multifaceted activity of MIF.

However, several aspects have to be clarified before the manuscript can be published.

Major

The sequence of chapters should be modified. The molecular aspect is of paramount importance for the IJMS readers, so the mechanisms of MIF activity in kidney diseases should appear after introduction, then clinical background of selected kidney diseases should follow.

The interpretation of MIF behaviour in the conditions of AKI seems unequivocal.

Apart from inflammatory background, one cannot forget that any condition decreasing GFR and urinary output may cause molecule accumulation in serum. Additionally, tubular damage may unable efficient reabsorption, thus increasing molecule content in urine. Could not this be one of the reasons for high MIF concentrations in AKI ? The same aspect should be discussed in the case of chronic kidney disease – CKD with its eGFR decrease may trigger accumulation of MIF as well.

Figure 1 contains information on MIF engagement in acute and chronic processes connected to obstructive nephropathy, but the text lacks it. The data on MIF in post-renal AKI and CAKUT-related CKD should be supplemented.

Acute tubular necrosis is not the only mechanism responsible for AKI and inflammation is not the sole driving force for AKI.

Experimental studies use the model of ischemia-reperfusion injury (IRI) to picture mechanisms accompanying AKI, but the statement about IRI-induced AKI suggests the authors do not distinguish between the cause and effect.

The chapter on the role of MIF inhibitors and anti-MIF antibodies should rather state about perspectives than therapies, because none of these agents have been used in humans.

Conclusions – the first paragraph would benefit from an attempt to summarize and interpret the multiplicity of MIF function in different kidney diseases and its molecular background.

The second paragraph should be moved to the section of perspectives – it is unusual to put new ideas and references to the conclusions.

Minor

English editing is recommended – the text contains spelling mistakes that make its understanding difficult, for example:

Line 36 - though  - should be through,

Line 70 – delectation – should be deletion,

Line 354 – bossibility – should be possibility,  etc.

Author Response

Major

The sequence of chapters should be modified. The molecular aspect is of paramount importance for the IJMS readers, so the mechanisms of MIF activity in kidney diseases should appear after introduction, then clinical background of selected kidney diseases should follow.

Response: We would like to thank the reviewer 2 for this very thoughtful suggestion to rearrange the manuscript with the mechanisms of MIF first followed by the clinical background and the role of MIF in various type of kidney diseases. However, we found that such changes seem to disrupt the flow of manuscript. Thus, we would like to keep the writing style from the clinical evidence for MIF in kidney diseases including AKI and CKD, followed by the role of MIF in kidney diseases, then the mechanisms of MIF, and completed it with the therapeutic approaches. It seems that this flow reads well and connects each other throughout the manuscript. We wish such arrangements are acceptable.   

The interpretation of MIF behaviour in the conditions of AKI seems unequivocal.

Response: We totally agreed with the reviewer 2 comments.  Indeed, there are diverse roles of MIF in AKI but so far no good reasons or interpretations are achieved. Why can MIF exert  pathogenic or protective in AKI? This question remains unexplained. To address the reviewer 2 concern, we generate a new scheme Figure 2 to illustrate the diverse roles of MIF in tubular inflammation vs proliferation. It is possible that high levels of MIF may trigger severe renal inflammation which may be pathogenic in AKI. Whereas MIF may also play a reparative role in AKI by promoting tubular cell proliferation while inhibiting apoptosis or cell cycle arrest if MIF levels are not high enough to trigger severe renal inflammation. Under this situation, MIF may be protective in AKI as demonstrated in recent studies that mice lacking MIF develop worse AKI by inhibiting tubular epithelial cell proliferation [28, 36, 37].  We have included such information in the New Figure 2 and discussed in the revised manuscript in Page 3, lines 86-107.

Apart from inflammatory background, one cannot forget that any condition decreasing GFR and urinary output may cause molecule accumulation in serum. Additionally, tubular damage may unable efficient reabsorption, thus increasing molecule content in urine. Could not this be one of the reasons for high MIF concentrations in AKI ? The same aspect should be discussed in the case of chronic kidney disease – CKD with its eGFR decrease may trigger accumulation of MIF as well.

Response: The points raised by the reviewer 2 is well taken. We have revised the manuscript by including these points as shown in Page 2, Lines 58-65.

Figure 1 contains information on MIF engagement in acute and chronic processes connected to obstructive nephropathy, but the text lacks it. The data on MIF in post-renal AKI and CAKUT-related CKD should be supplemented.

Response: as suggested by the reviewer 2, Information shown in Figure 1 has been included in the Text. The post-renal obstruction and congenital anomalies of kidney and urinary tract (CAKUT) as a cause of AKI have also been included in the revised Figure 1 and in Text as shown in Page 2, Lines 52-54.

Acute tubular necrosis is not the only mechanism responsible for AKI and inflammation is not the sole driving force for AKI.

Response: We totally agreed with the comment raised by the reviewer and the point has been included in the revised manuscript as shown in Page 2, Lines 58-65.

Experimental studies use the model of ischemia-reperfusion injury (IRI) to picture mechanisms accompanying AKI, but the statement about IRI-induced AKI suggests the authors do not distinguish between the cause and effect.

Response: Thank you for pointing it out the cause and effect of IRI-induced AKI, which has been stated clearly in the revised manuscript.

The chapter on the role of MIF inhibitors and anti-MIF antibodies should rather state about perspectives than therapies, because none of these agents have been used in humans.

Response: This chapter contains many experimental studies. These experimental therapeutic approaches are difficult to be include in the Conclusion and Perspectives section. However, the point raised by the reviewer 2 is well taken. To make this section more meaningful, we have now included a new Figure 5 to outline the various therapeutic approached. Following the reviewer’s suggestion, the limitations of these therapeutic approaches are also included in the Perspectives section in Page 10, Lines 445-447 

Conclusions – the first paragraph would benefit from an attempt to summarize and interpret the multiplicity of MIF function in different kidney diseases and its molecular background.

The second paragraph should be moved to the section of perspectives – it is unusual to put new ideas and references to the conclusions.

Response: The comments from the reviewer 2 are very helpful. We have revised this section as suggested by the reviewer 2. All revised sentences are highlighted in RED front in the revised manuscript (Page 10, Section 8). 

Minor

English editing is recommended – the text contains spelling mistakes that make its understanding difficult, for example:

Line 36 - though  - should be through,

Line 70 – delectation – should be deletion,

Line 354 – bossibility – should be possibility,  etc.

Response: Again, we would like to thank the reviewer 2 for so carefully reading through the manuscript and pointing out these typos. We have now carefully revised the manuscript and correct the English errors as much as we can.

Round 2

Reviewer 2 Report

I have no further comments

Author Response

Thank you very much for the the comment from the reviewer 2.

The revised manuscript has been checked for the spelling and typos.